# Etiology and Epidemiology of Croup before and throughout the COVID-19 Pandemic, 2018–2022, South Korea

**DOI:** 10.3390/children9101542

**Published:** 2022-10-09

**Authors:** Joon Kee Lee, Seung Ha Song, Bin Ahn, Ki Wook Yun, Eun Hwa Choi

**Affiliations:** 1Department of Pediatrics, Chungbuk National University Hospital, Chungbuk National University College of Medicine, Cheongju 28644, Korea; 2Department of Pediatrics, Seoul National University Children’s Hospital, Seoul 03080, Korea; 3Department of Pediatrics, Seoul National University College of Medicine, Seoul 03080, Korea

**Keywords:** child, COVID-19, croup

## Abstract

Omicron, a recent variant of severe acute respiratory syndrome coronavirus 2 (SARS-CoV-2), is currently globally dominating. We reviewed the etiology and epidemiology of croup over an approximately 5-year period, with an emphasis on the recent dominance of the Omicron variant. Children less than 5 years of age seen in the emergency department with diagnosis of croup from two large national tertiary hospitals were collected for the period from January 2018 through March 2022. Viral etiologies of the patients were compared with national surveillance data upon circulating respiratory viruses in the community. A total number of 879 croup cases were recognized during the study period. The most common pathogen was SARS-CoV-2 (26.9%), followed by HRV (23.8%), PIV1 (14.6%), PIV3 (13.1%), and CoV NL63 (13.1%), among seventeen respiratory viral pathogens tested by polymerase chain reaction. The viral identification rate was significantly higher in the Omicron period, with most of the pathogens identified as SARS-CoV-2. In the Omicron period, with the exponential increase in the number of COVID-19 cases in the community, croup associated with SARS-CoV-2 significantly increased, with a high detection rate of 97.2% (35 of 36) among croup cases with pathogen identified. The etiologic and epidemiologic data before and throughout the COVID-19 pandemic indicate that the association between croup and infection with the SARS-CoV-2 Omicron variant is highly plausible.

## 1. Introduction

The coronavirus disease 2019 (COVID-19) pandemic, which began in late 2019 to early 2020, is ongoing. Several variants of concern (VOCs) have appeared and disappeared. In general, both the transmissibility and the disease severity have shown to increase during the evolution of the severe acute respiratory syndrome coronavirus 2 (SARS-CoV-2) through Alpha to Delta VOC [1,2,3]. However, the clinical significance in children and adolescents has remained stably low compared to that in adults and elderlies [4]. In addition, asymptomatic infection is common in children [5]. The reason for this is still not clearly explained, but recent exposure to other viruses and routine vaccines—including but not limited to Bacillus Calmette–Guérin (BCG), measles-mumps-rubella (MMR), and tetanus-diphtheria-pertussis (Tdap) in children—are presumed to be associated with protective cross-reactive antibodies and T cells against SARS-CoV-2, based on a recent study [6].

Omicron, a recent variant of SARS-CoV-2, first reported from Botswana and very soon thereafter from South Africa in November 2021, is currently globally dominating [7]. Even though the transmissibility of the Omicron variant has increased via replication advantage and immune evasion, the preference for settling in the upper respiratory tract (URT) by the Omicron variant has probably reduced its severity in adults [8,9]. A recent study showed that SARS-CoV-2 is evolving toward more efficient aerosol generation and loose-fitting masks provide significant but only modest source control [10]. This may imply that the strength of viral replication in the intra-thoracic space may still be important. In any event, as young children have relatively small URT passages, more serious conditions can be expected in the pediatric population [11].

Croup is a common presentation of acute viral respiratory illness in children [12]. It is characterized by inspiratory stridor, barking cough, and hoarseness, which result from inflammation in the larynx and subglottic airway [13]. In a retrospective review characterizing cases of croup presenting to emergency departments (EDs) in the United States, approximately 1.3% of ED cases were due to croup, even though seasonal variations were prominent [12]. As the Omicron variant prevailed, croup patients were more frequently observed compared to periods before, according to the authors of this study. Furthermore, the literature on the association between the Omicron variant and croup in children has been increasing [14,15,16,17]. In consideration of the study that showed that the Omicron variant replicates faster than all other SARS-CoV-2 variants studied in the bronchi but less efficiently in the lung parenchyma, the association was even more plausible [9]. However, longitudinal observational studies, which include the era prior to the COVID-19 pandemic, are particularly limited. Studies with an extended observation period may reveal the association between croup in children and the Omicron variant with enhanced clarity. Therefore, we reviewed the etiology and epidemiology of croup over an approximately 5-year period, with an emphasis on the recent dominance of the Omicron variant.

## 2. Materials and Methods

### 2.1. Eligibility Criteria

The data of children less than 5 years of age seen in the ED with a recorded International Classification of Diseases, 10th revision (ICD-10), code for croup (J05.0), from January 2018 through March 2022, were collected from two major tertiary hospitals in South Korea: Chungbuk National University Hospital (CNUH, Cheongju) and Seoul National University Children’s Hospital (SNUCH, Seoul). The diagnosis of croup was made clinically by the attending physician in the ED, based upon the characteristic barking cough and stridor. Additional studies, such as radiographs or laboratory tests, were not required in making the diagnosis, even though ancillary tests were performed for selected patients.

### 2.2. Exclusion Criteria

Immunocompromised children and children with chronic pulmonary disease were excluded because the underlying conditions could affect the clinical appearance, making it difficult to provide an exact diagnosis of croup.

### 2.3. Sample Collection and Processing

Nasopharyngeal swab was obtained for viral detection from those who visited the ED with acute respiratory symptoms compatible with croup as a component of standard patient care. Because the collection was completed upon ED presentation, relation to symptom onset was different among patients. However, where the urgency of the croup symptoms is considered, the majority of patients would have presented right after the symptom onset. The samples were processed for viral detection as they were collected over time.

### 2.4. Respiratory Virus Identification

Data on basic demographics, respiratory virus detection by multiplex reverse transcription polymerase chain reaction (mRT-PCR), and/or SARS-CoV-2 detection by PCR at the time of croup diagnosis were retrospectively collected. The viruses included were as follows: adenovirus (AdV), influenza virus (IFV) A/B, parainfluenza virus (PIV) 1/2/3/4, human rhinovirus (HRV), human bocavirus (HBoV), coronavirus (CoV) 229E/NL63/OC43, enterovirus (HEV), metapneumovirus (MPV), and respiratory syncytial virus (RSV) A/B. SARS-CoV-2 was detected by commercial mRT-PCR kits: AccuPower^®^ COVID-19 Multiplex Real-Time RT-PCR (Bioneer, Inc., Daejeon, Korea) in CNUH and STANDARD™ M nCoV Real-Time-Detection (SD Biosensor, Inc., Suwon, Korea) in SNUCH. Other respiratory viruses were detected with Anyplex™ II RV16 Detection (Seegene, Inc., Seoul, Korea) in both hospitals. Respiratory viruses with epidemiological associations with croup were selected for further investigation. Information regarding circulating respiratory viruses in the community was collected from the Laboratory Surveillance Service for influenza and respiratory viruses, which is monitored by the Korea Disease Control and Prevention Agency (https://www.kdca.go.kr/ accessed on 12 September 2022). The viral identification rate was defined as the proportion of patients with identified viral etiology among all patients diagnosed with croup, which included patients who did not undergo testing at all.

### 2.5. Study Period Division

The study period was divided into three subperiods: pre-COVID-19 (January 2018 to 3rd week of 2020), pre-Omicron (4th week of 2020 to 3rd week of 2022), and Omicron (4th week of 2022 to March 2022). The beginning of the pre-Omicron and Omicron periods was the week in which the first SARS-CoV-2 infection was confirmed in a patient and the week in which 50% of the monitored SARS-CoV-2 variants were identified as Omicron in South Korea, respectively.

### 2.6. Statistical Analyses

Statistical analyses were performed using SPSS Statistics for Windows version 26.0 (IBM Corp., Armonk, NY, USA). Categorical data were analyzed using the chi-squared test or Fisher’s exact test, as appropriate. The linear-by-linear association model was used for trend analyses. *p* < 0.05 was considered statistically significant.

### 2.7. Ethical Consideration

This study was approved by the Institutional Review Boards of Chungbuk National University Hospital (No. 2022-04-007) and Seoul National University Children’s Hospital (No. 1102-084-353).

## 3. Results

### 3.1. Case Numbers and Demographics

A total of 879 croup cases were recognized during the study period. The median age of the patients was 22 months (range 2–59 months), and 64.7% of the patients were male (Table 1). The case numbers showed significant differences among the study periods. The case numbers of the pre-COVID-19 era and the pre-Omicron era were 670 and 145, respectively. The case number of the Omicron era was 64. However, as the study period of the Omicron era was comparably shorter than the prior two eras, direct comparison was not feasible. Therefore, the number of cases/week for each era was calculated. For the whole study period, 3.99 cases/week was recorded. However, in the pre-Omicron era, this rate decreased to 1.39 cases/week. Finally, the highest number of cases per week was demonstrated in the Omicron era: 7.11 cases/week.

### 3.2. Viral Pathogens

In 130 out of 879 (14.8%) cases, viral pathogens were identified. The most common pathogen was SARS-CoV-2 (26.9%), followed by HRV (23.8%), PIV1 (14.6%), PIV3 (13.1%), and CoV NL63 (13.1%) (Table 2). HRV was commonly the most common pathogen both in the pre-COVID-19 era and the pre-Omicron era (Figure 1). In both eras, besides HRV, PIVs were the major pathogen responsible for croup. However, in the Omicron era, the main viral pathogen identified was SARS-CoV-2.

### 3.3. Multiple Respiratory Virus Detections

Multiple viruses were identified in 26 out of 130 (20.0%) cases, with 2 and 3 viruses detected in 21 and 5 cases, respectively (Table 3). Among the cases with codetection, the most common virus was HRV (29.8%), followed by PIV1 (14.0%), and AdV (12.3%). Even when the cases were divided into dual and triple codetections, there was no difference in the most common pathogen (HRV) responsible for croup. However, AdV and HBoV were more common in cases with triple detections compared to those in dual detections. None of the croup cases with SARS-CoV-2 had codetection.

### 3.4. Viral Identification Rate and Number of Tests Performed

The viral identification rate was significantly higher in the Omicron period (56.3%, *p* < 0.001) than in the pre-COVID-19 (12.4%) and pre-Omicron (7.6%) periods. However, the proportion of patients going through mRT-PCR tests for respiratory viruses other than SARS-CoV-2 decreased from 14.0% (pre-COVID-19) to 1.5% (Omicron) throughout the study period (*p* = 0.001). The proportion of PCR tests for SARS-CoV-2 increased from 40.4% (pre-Omicron) to 84.4% (Omicron) (*p* < 0.001).

### 3.5. Epidemiological Association

Based on both the knowledge of common etiologies of croup and the etiologic results of the current study, PIV and CoV were selected for further analysis [18,19,20]. The cases in this study were compared to the national surveillance data (Figure 2). In general, each croup epidemic (five or more cases of croup per week in clusters) was associated with a high detection rate of PIV or CoV. The number of cases of croup remained very low during the pre-Omicron period. In the Omicron period, with the exponential increase in the number of COVID-19 cases in the community, croup caused by SARS-CoV-2 infection significantly increased (7.11 cases/week), with a high detection rate of 97.2% (35 of 36) among croup cases with pathogen identified.

## 4. Discussion

By investigating the etiology and epidemiology of croup in both the periods before and throughout the COVID-19 pandemic, the association of croup with SARS-CoV-2 Omicron infection is suggested. As the etiologies of croup investigated in the study period prior to the COVID-19 pandemic is correlated with known patterns, we believe that the findings discovered in this study are reliable.

The impact of the COVID-19 pandemic has been crucial to all aspects of human life. This did not leave out children and adolescents. A certain extent of relief remained as medical needs in the pediatric population was relatively lower than those of the adults and elderlies. However, the children were one of the most vulnerable groups in the pandemic for several reasons, including but not limited to psychosocial harmful influences [21]. However, in terms of medical care, SARS-CoV-2 may have not been a significant pathogen in the pediatric population. Furthermore, even with evolutions in VOCs, children probably have remained in the safety zone in terms of demanding medical care. This seems to have totally changed due to the Omicron variant.

Several reports on croup associated with the Omicron variant have been published [14,15,22]. However, in general, the study period included in the previous studies was limited to the COVID-19 era. In contrast, the present analysis includes a comparably long period, including the pre-COVID-19 era, and provides highly reliable findings by demonstrating the usefulness and reliability of the methodology used in the current study. The observations revealed that despite extensive SARS-CoV-2 surveillance in South Korea, during the pre-Omicron period, no single croup case due to infection with SARS-CoV-2 was reported. However, after the rapid spread of Omicron, without the spread of PIV or other CoVs, croup cases due to infection with SARS-CoV-2 showed a noticeable increase. Furthermore, even with the substantial increase in the number of COVID-19 cases in late 2021 (pre-Omicron), a croup associated with SARS-CoV-2 infection was not noted. These observations possibly verify the association between infection with the Omicron variant and croup in children, which is consistent with the results of prior publications [14,22].

Another alarming situation in dealing with the Omicron variant may be its high transmissibility, especially in children. In South Korea, prior to the Omicron era, even with several epidemic waves, the pediatric population was not the major group infected with SARS-CoV-2. Even though this epidemic wave introduced by the Omicron variant affected all age groups, the Omicron variant caused an exponential increase of infectees in the pediatric population. This is partly explained by the replication advantage of the Omicron variant. The increased household transmission rates of VOCs have been shown in recent studies [23,24,25]. Therefore, it is probable that future VOCs may have enhanced transmissibility, especially in the pediatric group. As has been the case with the Omicron variant, we cannot foresee diseases associated with future VOCs. Enhanced monitoring in all aspects of managing children affected by COVID-19 is in demand.

Croup is a viral disease and the etiology of the disease has interested a lot of researchers. Interestingly, the study results show that ‘any’ virus could cause croup. However, it is well-recognized that PIVs are the main cause of the majority of types of croup. Among PIVs, PIV1 and PIV3 are known to account for a large portion [18], which is consistent in the current study. We find that the association of the Omicron variant with croup is comparable with the discovery of a new coronavirus subtype, which was also closely related with croup. CoV NL63 was first identified in the early 2000s [26]. As the virus was first discovered in a 7-month-old child suffering from bronchiolitis and conjunctivitis, the association with croup may not have been assumed at first. However, after several observations, the association with croup was relatively clear [19,20,27]. In this aspect, it is quite interesting that only certain subtypes, such as PIV1/3, CoV NL63, or the Omicron variant of SARS-CoV-2, are associated with croup. We believe that future studies focusing on specific coronavirus types, including but not limited to SARS-CoV-1/2, are needed, which would elucidate the reason for this phenomenon.

There were several cases of codetected viruses in the cases of the current study. Furthermore, a large portion of the viruses detected were HRV, which may cause confusion in the analysis of the etiology of croup. Due to the ubiquity and the questioned role of HRV in croup, even though it was the second most commonly identified virus for the total study period, we cannot conclude that HRV was the true pathogen [28]. HRV is the etiologic agent of most common colds and children are the major reservoir for HRV [29]. Therefore, a large portion may have been accidentally detected. However, on the other hand, HRV is known to be symptomatic in most young children [30]. Considering these conflicting aspects of the influence of HRV in the disease, we are not sure of the true clinical significance of HRV detected in this study. In terms of coinfection or codetection, certain studies report that coinfection with rhinovirus increases the risk of lower respiratory tract infection [28]. However, the role of coinfection with HRV in croup has not been fully investigated. We believe that further studies will elucidate the questions raised.

The viral identification rate among croup cases in this study seems to be comparably low (14.8%). As identification of the viral pathogen is generally not a component of the management of croup, this is not surprising [31]. Diagnostic testing for SARS-CoV-2 was frequently performed for those who have any respiratory symptoms due to the ‘testing and tracing’ strategy implemented during the COVID-19 pandemic in South Korea. In contrast, further mRT-PCR testing to detect respiratory viruses was rarely performed after the SARS-CoV-2 PCR testing initially tested positive. However, as shown by the cases/week variables, we assume that the increase of croup cases in the Omicron era is truly associated with the Omicron variant of SARS-CoV-2. In addition to less mRT-PCR testing, large-scale SARS-CoV-2 testing led to a lower viral identification rate among children with croup in the pre-Omicron period compared to the pre-COVID-19 period. The viral identification rate was higher in the Omicron period than in the pre-Omicron period due to the high detection of SARS-CoV-2 associated with croup in the Omicron period. In terms of viral codetection, as there was no case of codetection in the Omicron period, the attribution of other viruses in the Omicron period is not clear. Because of the lack of experience with SARS-CoV-2, the definite causal attribution of the virus to croup (similar to HRV) is also probable but not yet certain. Further studies, including in vitro studies, may elucidate the mechanism of this association if it exists.

### Limitations

There are a few limitations to this study. As mentioned above, the viral causes of the cases in the current study seemed to account for a comparably low portion compared to those in national surveillance data. However, without a prospective study design to test all croup patients for viral pathogens, the situation is assumed to be similar in most health care centers [31]. The absence of patient-level outcomes and clinical trajectories in the current study remains a considerable limitation. We believe that these data might be improved upon in a single-center study with consistent policies in the management of croup. In addition, we did not include data upon bacterial identification, due the nature of the viral origin of croup. However, there could be important relationships between the microbiomes that mediate viral infection, so the coinfection information may be useful if collected [32].

## 5. Conclusions

In conclusion, the etiologic and epidemiologic data before and throughout the COVID-19 pandemic indicate that the association between croup and infection with the SARS-CoV-2 Omicron variant is plausible.

## Figures and Tables

**Figure 1 children-09-01542-f001:**
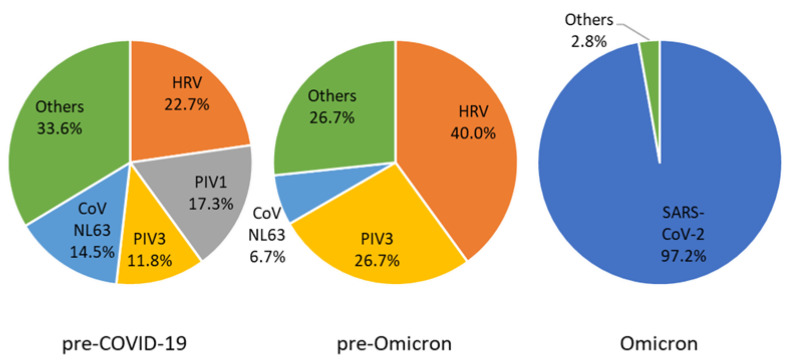
Proportional distribution of viral pathogen in children responsible for croup during 2018–2022, South Korea. PIV, parainfluenza virus; HRV, human rhinovirus; CoV, coronavirus; SARS-CoV-2, severe acute respiratory syndrome coronavirus 2.

**Figure 2 children-09-01542-f002:**
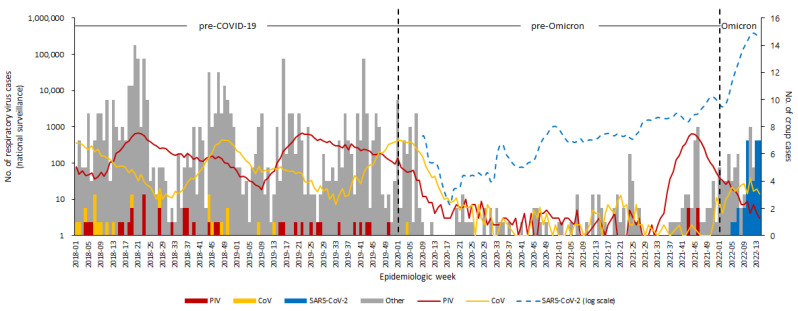
Epidemiologic curve of croup patients from two tertiary hospitals with surveillance data of parainfluenza viruses and coronaviruses, including severe acute respiratory syndrome coronavirus 2 circulating in the community, 2018–2022, South Korea. The broken line indicates circulating respiratory viruses in the community collected from the Laboratory Surveillance Service (Korea Disease Control and Prevention Agency, https://www.kdca.go.kr/ accessed on 12 September 2022). The blue dashed line indicates the mean numbers of weekly SARS-CoV-2 confirmed cases (Korean Ministry of Health and Welfare, http://ncov.mohw.go.kr/en, accessed on 1 August 2022). All lines correspond to the y-axis on the left. Each bar shows croup cases from study sites. Croup cases with detection of parainfluenza virus or coronavirus are correspondently colored. The gray bar indicates croup cases with other or unknown etiology. Each bar corresponds to the y-axis on the right. The left y-axis corresponding to viral surveillance data is presented on a logarithmic scale. COVID-19, coronavirus; PIV, parainfluenza virus; CoV, coronavirus; SARS-CoV-2, severe acute respiratory syndrome coronavirus 2.

**Table 1 children-09-01542-t001:** Demographics of children diagnosed with croup during 2018–2022, South Korea.

Characteristic	Total	Pre-COVID-19 (Jan 2018–3rd Week 2020)	Pre-Omicron (4th Week 2020–3rd Week 2022)	Omicron (4th Week 2022–Mar 2022)
Total weeks	220	107	104	9
Case number	879	670	145	64
Cases/week	3.99	6.26	1.39	7.11
Sex (male)	569 (64.7)	433 (64.6)	97 (66.9)	39 (60.9)
Median age in month (range)	22 (2–59)	22 (2–59)	24 (2–58)	16.5 (3–54)
mRT-PCR-tested	106 (12.1)	94 (14.0)	11 (7.6)	1 (1.5)
SARS-CoV-2 PCR-tested	112 (12.7)	NA	58 (40.0)	54 (84.4)
Viral identification	130 (14.8)	83 (12.4)	11 (7.6)	36 (56.3)
Codetection	26 (20.0)	23 (27.7)	3 (27.3)	0 (0)

Values are shown in No. (%) unless stated otherwise. COVID-19, coronavirus disease 2019; mRT-PCR, multiplex reverse transcription polymerase chain reaction; SARS-CoV-2, severe acute respiratory syndrome coronavirus 2; NA, not available.

**Table 2 children-09-01542-t002:** Virologic distributions in children diagnosed with croup during 2018–2022, South Korea.

Virus	Total	Pre-COVID-19 (Jan 2018–3rd Week 2020)	Pre-Omicron (4th Week 2020–3rd Week 2022)	Omicron (4th Week 2022–Mar 2022)
SARS-CoV-2	35 (26.9)	NA	0 (0)	35 (97.2)
HRV	31 (23.8)	25 (30.1)	6 (54.5)	
PIV1	19 (14.6)	19 (22.9)		
PIV3	17 (13.1)	13 (15.7)	4 (36.4)	
CoV NL63	17 (13.1)	16 (19.3)	1 (9.1)	
AdV	8 (6.2)	8 (9.6)		
HBoV	8 (6.2)	5 (6.0)	3 (27.3)	
CoV OC43	5 (3.8)	5 (6.0)		
RSV A	5 (3.8)	5 (6.0)		
RSV B	5 (3.8)	3 (3.6)	1 (9.1)	1 (2.8)
Flu B	3 (2.3)	3 (3.6)		
PIV2	3 (2.3)	3 (3.6)		
MPV	3 (2.3)	3 (3.6)		
Flu A	1 (0.8)	1 (1.2)		
HEV	1 (0.8)	1 (1.2)		
Total	161 (100)	110 (100)	15 (100)	36 (100)

Values are shown in No. (%). COVID-19, coronavirus disease 2019; Flu, influenza virus; AdV, adenovirus; HEV, enterovirus; PIV, parainfluenza virus; MPV, metapneumovirus; HBoV, human bocavirus; HRV, human rhinovirus; CoV, coronavirus; RSV, respiratory syncytial virus; SARS-CoV-2, severe acute respiratory syndrome coronavirus 2.

**Table 3 children-09-01542-t003:** Respiratory virus codetection in children diagnosed with croup during 2018–2022, South Korea.

Characteristic	Total	Dual Detection	Triple Detection
HRV	17 (29.8)	13 (31.0)	4 (26.7)
PIV1	8 (14.0)	7 (16.7)	1 (6.7)
AdV	7 (12.3)	4 (9.5)	3 (20.0)
CoV NL63	6 (10.5)	5 (11.9)	1 (6.7)
HBoV	5 (8.8)	2 (4.8)	3 (20.0)
PIV3	4 (7.0)	3 (7.1)	1 (6.7)
RSV A	3 (5.3)	2 (4.8)	1 (6.7)
MPV	2 (3.5)	2 (4.8)	
CoV OC43	2 (3.5)	1 (2.4)	1 (6.7)
Flu A	1 (1.8)	1 (2.4)	
HEV	1 (1.8)	1 (2.4)	
RSV B	1 (1.8)	1 (2.4)	

Values are shown in No. (%). Flu, influenza virus; AdV, adenovirus; HEV, enterovirus; PIV, parainfluenza virus; MPV, metapneumovirus; HBoV, human bocavirus; HRV, human rhinovirus; CoV, coronavirus; RSV, respiratory syncytial virus; mRT-PCR, multiplex reverse transcription polymerase chain reaction; SARS-CoV-2, severe acute respiratory syndrome coronavirus 2.

## Data Availability

The datasets used and/or analyzed during the current study are available from the corresponding author on reasonable request.

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
