# Peer review of "Etiology and Epidemiology of Croup before and throughout the COVID-19 Pandemic, 2018–2022, South Korea"

_children, 2022, doi:10.3390/children9101542_

Round 1
Reviewer 1 Report
Children article review
Etiology and Epidemiology of Croup before and throughout the COVID-19 Pandemic, 2018-2022, South Korea
Abstract:
Line 21: consider saying “croup associated with SARS-CoV-2 detection reached epidemic proportions”?
It may also help to briefly indicate how many pathogens you probed for — this would help support the idea that you likely detected the causative agent.
Line 22: “high rate of detection” — how high? Or up to how high? Could help to quantify with a number/range/confidence interval/etc. here.
Intro:
Page 1 line35: which routine vaccines in particular?
Page 1 line 42: consider mentioning research from Don Milton’s group that showed that omicron and delta both evolved to have potential for higher viral shedding into fine aerosols. This could mean that the strength of viral replication in the intra-thoracic space may still be important.
Page 2 line 48: check grammar — sentence seems off or is unclear.
I’d like to see more said about why in particular you wanted to look at the association of croup with SARS-2? For example, did the hypothesis derive from observations of elevated croup among pediatric group? Among acute respiratory illnesses in children, how common is croup compared with other symptoms? It would also be helpful to describe refs 8 and 9 — what did they do and what did they find regarding omicron replication in the URT? How does this inform how you decided to look at croup in the pediatric population? Perhaps not much was known and so this was more of any exploratory mission. In any case, would be helpful to clarify briefly in the intro.
Methods:
Page 2 line 67: If you can provide more information about primers/probes or kits used, this might be helpful for others trying to do similar work and for reproducibility?
I think it is necessary to describe how the clinical samples were collected, and also when they were collected in relation to symptom onset. For example — are these all nasopharyngeal swab samples? Anterior nares? Midturminate swabs? Saliva? Etc? A combination?
Were these samples all collected and banked and run later on or were they run as they were collected over time? Is there reason to believe that there would be more sample collection and testing during the covid period because of concerns about the pandemic virus?
Results:
Table 1: might help to add a row describing the number of total weeks included in the surveillance for each study period? Could help to put these case numbers as a rate over the study period to normalize by time. This would, in fact, make comparisons between the study periods more feasible and then you could do some statistics, e.g., rate ratios, etc.
Why not stratify these results by virus or bacteria detected and present all the data for each virus probed for on the multiplex assay? Alternatively, this could go in a supplemental table, or a figure. Visually you could probably show very eloquently the rise in SARS-2 associated croup cases from periods 1 to 3. I see that you did this for the viruses in Table 2, but it might also be interesting to see the bacteria presented in this way, if not in the main text, then in the supplement.
Did the viral coinfected have more croup?
Table 3: Please consider extended this analysis to include the bacteria or use a separate table. There could be important relationships between microbiome that mediate viral infection, so the confection information would be useful to share.
Page 5 line 143: Is this talking about the proportion of positive tests? I think so, and it might help to clarify for readability.
Page 5 line 150: typo with the word “five”?
Page 5 line 154: what is your definition of epidemic proportions and “high rate of detection”?
Figure 2: Please clearly state that the red and yellow lines correspond with the y-axis on the left? I think this is correct? I had to assume based on the info in the legend.
Did you try to run any statistical tests to show relationship between croup and SARS-2 detection during the omicron wave? Or I missed it?
Discussion:
Page 7 line 218: What types of studies would you suggest for further investigation?
Page 9 line 234: Would be interesting to hear more about this low detection of viruses. What about bacteria? What other explanations could there be for this and what is generally thought about the etiology of croup if not associated with viral detection? Could it be that croup symptoms onset occur after infection is cleared, or prior to viral detection in clinical swabs?
Limitation/area for future work: exclusion of immunocompromised children and/or children with chronic pulmonary disease — how are these children experiencing viral infections and symptoms, especially during the omicron wave and in comparison with other pediatric groups? Could be for future work?
Reviewer 2 Report
Overall, the level of English seems to be good, as far as I can judge as a non-native speaker. Since I found only isolated small errors (line 196 maybe - may be, line 202 have shown - have been shown) I would recommend to have a native speaker proofread.
As you say yourself, "any" virus can cause croup. Therefore, I think it is important to describe how the diagnosis of croup is made, since you have not evaluated any diagnostic methods, but "only" the ICD coding.
I think it is important to emphasize that because of the lack of experience with SARS-CoV-2, the definite causal attribution of the virus to croup (similar to HRV) is also probably not yet certain (vs. coinfection/codetection). Your findings that no other virus was detected in the SARS-CoV-2 cases support this, but I would at least discuss it.
I find the point you discuss between lines 233 and 244 very important and would pay even more attention to it, unfortunately also in relation to the limitations of the study. Once SARS-CoV-2 has been detected in routine screening, it is likely that no further PCR was performed because they were satisfied with the one positive result. You should still emphasize the different rates of the pre-Omicron and Omicron era here to support your point of the importance of SARS-CoV-2 for croup.
Round 2
Reviewer 1 Report
I approve of the revisions and thank the authors for their thoughtful replies. I vote to accept the revised manuscript.